# From Laboratory to Field: Concurrent Validity of Kinovea’s Linear Kinematics Tracking Tool for Semi-Automated Countermovement Jump Analysis

**DOI:** 10.3390/s26010024

**Published:** 2025-12-19

**Authors:** Lucija Faj, Jelena Aleksić, Olivera M. Knežević, Branislav Božović, Hrvoje Brkić, Damir Sekulić, Dragan M. Mirkov

**Affiliations:** 1Faculty of Kinesiology, University of Osijek, 31000 Osijek, Croatia; 2Faculty of Sport and Physical Education, University of Belgrade, Blagoja Parovica 156, 11000 Belgrade, Serbia; jelena.aleksic@fsfv.bg.ac.rs (J.A.);; 3Faculty of Medicine, University of Osijek, 31000 Osijek, Croatia; 4Faculty of Dental Medicine and Health, University of Osijek, 31000 Osijek, Croatia; 5Faculty of Kinesiology, University of Split, 21000 Split, Croatia; dado@kifst.hr

**Keywords:** markerless motion capture, video analysis, vertical jump performance, sports biomechanics, performance monitoring

## Abstract

Affordable high-frame-rate cameras and open-source software, such as Kinovea (ver. 2025.1.0), have expanded the potential for conducting kinematic assessments outside laboratory settings. This study examined the reliability and validity of Kinovea’s semi-automated linear kinematics tracking tool by comparing its outputs with those from a 3D marker-based motion capture system (Qualisys). Ten recreationally active male basketball players (x̄ ± SD: age 23.7 ± 1.7 years; height 183 ± 5 cm; body mass 76.8 ± 9.8 kg) performed three CMJ trials, simultaneously recorded using both systems. Reflective markers placed on the shoulder, hip, and knee were tracked in Kinovea by two raters with different levels of experience to extract core CMJ variables (total take-off time and maximum vertical displacement) and complementary variables (eccentric and propulsion duration, and minimum vertical displacement). Inter-rater reliability and concurrent validity were evaluated using intraclass correlation coefficients (ICCs), coefficients of variation (CV%), standard error of measurement (SEM), and Bland–Altman analysis. Results showed excellent inter-rater reliability (ICC = 0.73–0.99) across all markers, with the hip and knee demonstrating the highest consistency. Strong validity relative to Qualisys was observed for both raters (ICC = 0.68–0.99; r > 0.80), with small systematic biases primarily in temporal variables. Collectively, these findings demonstrate that Kinovea’s semi-automated 2D analysis yields reliable and valid CMJ measurements comparable to 3D motion capture, even for less experienced users. As a free and easily deployable tool, it offers a widely accessible alternative for field-based performance monitoring and applied biomechanics research where laboratory-grade equipment is not available.

## 1. Introduction

The growing physical and technical demands of modern sport have made biomechanical evaluation of movement increasingly important for optimizing performance and minimizing injury risk [1]. Within this field, kinematic analysis examines the spatial and temporal characteristics of movement, such as displacement, velocity, and acceleration, which are crucial for understanding movement efficiency [2]. With advances in technology, motion analysis has become a central component of evidence-based approaches in sports training, rehabilitation, and performance monitoring [3].

Among the most commonly analyzed athletic movements, the countermovement jump (CMJ) is widely recognized as the fundamental testing modality for assessing lower-limb power and neuromuscular function [3,4]. Its simplicity, reliability, and sensitivity to changes in training status make it a standard diagnostic tool across sports science and clinical settings [5]. The take-off phase is particularly important, as it represents the point at which the athlete converts stored elastic energy and muscular force into vertical momentum, directly influencing the jump height achieved during flight [6,7]. Collectively, take-off time and jump height represent one of the core variables for analyzing CMJ performance, providing valuable insight into an athlete’s mechanical efficiency, explosive capacity, and neuromuscular performance [2]. For this reason, CMJ assessment is widely implemented in sports such as basketball, volleyball, and soccer, where jump performance underpins many technical actions [8,9]. It is equally applied in youth development and rehabilitation settings, where it serves as an indicator of motor learning, functional recovery, and movement quality [10].

However, the accessibility of accurate CMJ kinematic analysis remains limited by the reliance on laboratory-grade, marker-based 3D motion capture systems, which are recognized as the “gold standard” for biomechanical assessments [3,11]. These systems use multiple high-speed infrared cameras to capture the position of reflective markers in three-dimensional space with high accuracy, but their cost, technical demands, and restricted laboratory setup make them impractical for routine use in applied sport settings [12]. To address these limitations, 2D video analysis tools such as Kinovea [13], an open-source motion analysis software distributed under the GPLv2 license (Kinovea ver. 2025.1.0 for Windows; available at https://www.kinovea.org (accessed on 14 August 2025), have gained popularity among researchers and practitioners. Consequently, an important question has emerged regarding how accurately these tools can detect the fundamental temporal and kinematic events of a vertical jump.

Most studies analyzing vertical jumps with these tools have relied on manual frame-by-frame tracking to identify key movement events, including take-off and landing. These approaches have demonstrated excellent validity and reliability when compared with force plates or 3D motion capture systems, particularly for the measurement of jump height and flight time [14,15]. For instance, Balsalobre-Fernández et al. (2018) reported a near-perfect agreement (r = 0.99) between Kinovea-derived and force-platform jump heights [14], while Pueo et al. (2020) confirmed that both Kinovea and smartphone-based video analysis are reliable and useful tools for assessing vertical jump performance [15]. However, despite their practical advantages, manual frame-by-frame approaches are inherently limited by subjectivity in frame selection, reduced temporal precision, and potential parallax errors resulting from camera misalignment or out-of-plane movement [16].

To overcome the limitations of manual 2D analysis, Kinovea introduced a semi-automated linear kinematics tracking tool, which allows for continuous tracking of selected anatomical landmarks throughout the entire movement cycle. This function minimizes user bias, improves temporal resolution, and enables the extraction of complete displacement–time curves rather than isolated key frames. Consequently, the semi-automated approach provides a more objective method for quantifying countermovement jump (CMJ) kinematics compared to traditional manual 2D methods. However, its accuracy may still be influenced by factors such as marker placement, movement complexity, and recording conditions [17]. Notably, little evidence exists on how raters’ experience influences measurement accuracy, which represents an important gap in the literature that requires further investigation [15,17].

The purpose of this study was to evaluate the feasibility of a semi-automated 2D video analysis approach for assessing countermovement jump (CMJ) performance using Kinovea’s linear kinematics tracking tool. Specifically, the study aimed to (1) examine inter-rater reliability between raters with different levels of experience using the software, and (2) determine the concurrent validity of this semi-automated 2D method against a “gold standard” 3D marker-based motion capture system (Qualisys; Qualisys Medical AB, Gothenburg, Sweden) [18]. By leveraging the semi-automated tracking function to extract continuous displacement–time signals, this approach sought to provide a more feasible and accessible alternative to fully manual 2D analysis methods in both research and applied sports settings.

## 2. Methods

### 2.1. Experimental Session

The participants in this study were ten healthy, recreationally active male basketball players, defined as training at least three days per week for a minimum of 60 min per session (x̄ ± SD: age = 23.7 ± 1.7 years; height = 183 ± 5 cm; body mass = 76.8 ± 9.8 kg). All subjects had previous experience with the CMJ jump and reported no previous neuromuscular injuries that might compromise their performance. Written informed consent was obtained from each participant before the testing procedure, adhering to the Declaration of [19]. The testing protocol was approved by the Institutional Ethical Review Board of the University of Belgrade (Approval No. 02-848/23-2; 5 May 2023).

Sample size was estimated using G*Power (version 3.1.9.6, Heinrich-Heine University, Düsseldorf, Germany), based on a statistical power (1–β) of 0.8, significance level (α) of 0.05, and correlation of 0.8, resulting in a required minimum of 9 participants. The assumed correlation coefficient of 0.8 reflects effect sizes commonly reported in previous CMJ validation studies [14,15]. 

### 2.2. Experimental Protocol

After a standardized dynamic warm-up and 2–3 trial jumps, participants performed three CMJ trials with a 1 min rest interval between each trial to minimize fatigue. Each trial was simultaneously recorded using a 3D marker-based motion capture system (Qualisys Medical AB, Gothenburg, Sweden) and a smartphone camera (iPhone 13; Apple Inc., Cupertino, CA, USA).

Four Qualisys Miqus M3 infrared cameras were used to record full-body motion during all jump phases, capturing one side of the body (Figure 1). Cameras were set at 300 Hz to ensure high temporal resolution. Before each session, the Qualisys system was calibrated using a triangular reference object of known dimensions placed on the floor to establish the X and Y axes, and a handheld calibration wand was used to define the Z axis. The participants were instructed to place their feet at that point for standardized starting positions.

Reflective markers were unilaterally placed on four anatomical landmarks: shoulder (acromion), hip (greater trochanter), knee (lateral femoral condyle), and toe (5th metatarsal). While trajectories from the shoulder, hip, and knee markers were captured reliably, the toe marker exhibited frequent occlusions and increased signal noise across multiple trials, resulting in intermittent loss of data. These occlusions were primarily due to insufficient visual contrast between the marker and participants’ footwear, as most participants wore predominantly white shoes similar in color to the marker, which impaired reliable tracking in Kinovea. In contrast, clothing worn on proximal segments provided greater visual contrast relative to the markers, yielding more stable trajectories for the shoulder, hip, and knee. Retaining the toe marker would have required extensive manual, frame-by-frame adjustments and interpolation, which was intentionally avoided in order to demonstrate the feasibility of the underlying principle of semi-automated CMJ analysis. Given that the primary focus of this study was on global CMJ displacement and temporal characteristics rather than foot- or ankle-specific kinematics, only the shoulder, hip, and knee markers were retained for further analysis. This methodological choice does not compromise the reliability of the reported results.

The smartphone camera was positioned perpendicular to the ground, capturing the left sagittal plane at a distance of ~2 m. The camera tripod was set at ~1.10 m height (i.e., participant’s mid-trunk level) (Figure 1). Videos were recorded at 1080 p HD, 240 frames per second (fps), allowing for detailed motion analysis, as recommended for biomechanical applications [16]. Recordings were subsequently analyzed in Kinovea software by two raters, one experienced, with approximately two years of prior experience using the software for biomechanical video analyses [3], and one novice conducting measurements for the first time. Both raters were familiarized with the functionality of Kinovea’s linear kinematics tracking tool using the official software documentation and user guidelines [13]. The experienced rater had prior experience using this tool in previous analyses, whereas the less experienced rater had no prior hands-on experience with Kinovea’s linear kinematics feature beyond this familiarization phase.

Participants were requested to wear tight, darker-colored sports clothing to enhance body–background contrast for better segment tracking. All tests were conducted under standardized laboratory conditions for each participant.

### 2.3. Data Processing

Measurements were obtained using a 3D motion capture system (Qualisys) as the reference, and 2D Kinovea analysis was conducted by two raters with different levels of experience.

#### 2.3.1. Marker-Based Data

Marker-based data were exported using Qualisys Track Manager [18] to obtain three-dimensional positional data for each participant. A four-marker model was created for each participant, with reflective markers placed on the acromion, greater trochanter, lateral femoral condyle, and fifth metatarsal. This setup enabled accurate tracking of segmental movements during the CMJ. Missing trajectories caused by occlusion were reconstructed using linear or polynomial interpolation, while a moving average filter was applied to minimize high-frequency noise without altering the underlying motion pattern. It is important to note that interpolation was required only for brief gaps in the marker trajectories caused by momentary occlusions during dynamic movement phases. These gaps were limited to short durations, typically spanning only a few consecutive frames, and occurred in a small proportion of the recorded trajectories. No cases of prolonged signal loss were observed. This approach is consistent with standard practices in marker-based motion capture analysis, where brief interpolation is commonly used to address transient occlusions without substantially affecting measurement accuracy.

Subsequent analyses in MATLAB (ver. 25.1; MathWorks, Natick, MA, USA) involved tracking marker displacement over time and computing velocity profiles through numerical differentiation. The transition from the eccentric to concentric phase was identified based on changes in vertical displacement and toe marker contact, following established biomechanical analysis procedures [3,7].

#### 2.3.2. Kinovea Data

The Kinovea software required 2D calibration, which was performed relative to the participant’s height (Figure 2). The calibration procedure involved drawing two parallel horizontal lines, one positioned at the level of the feet and the other at the top of the head, followed by drawing a vertical reference line between them. This vertical line was then scaled according to the participant’s height, previously measured with an anthropometer.

Each video was analyzed independently by two raters, who manually identified each anatomical landmark on the participant based on the reflective markers placed on the body during data collection. After initial identification, raters applied the trajectory tracking (Track Path) function to automatically follow the markers throughout the movement. The tracking process in Kinovea was performed using a dual-window method, consisting of an object window (a small region centered on the marker) and a search window (a larger region used by the software to predict the marker’s location in the next frame). To avoid potential bias, each rater conducted the tracking independently, without access to the other’s results, and without prior knowledge of the 3D reference data. Tracking accuracy was visually inspected frame by frame, and manual adjustments were made whenever the automated algorithm lost precision, for example, due to motion blur or partial occlusion, following the procedures described in the official Kinovea documentation [13].

Vertical displacement plots were created for subsequent data processing (Figure 3). Position data were then exported using Kinovea’s Export to Spreadsheet function, providing raw coordinates for each marker over time. For more detailed analysis, including signal processing and further kinematic computations, the exported data were subsequently analyzed in MATLAB.

### 2.4. Variables

The variables selected for analysis (Table 1) were informed by prior research on vertical jump biomechanics [3,7,20,21].

The core variables included jump height from maximum vertical displacement (h_Z_max) and take-off duration (dt_take_off), both selected for their practical relevance to assessing vertical jump performance [4,6,7]. Maximum jump height was defined as the vertical displacement of the marker from the instant of toe-off to its highest point during the flight phase. Take-off duration represented the total time from the initiation of the downward movement to the instant of take-off, encompassing both the eccentric and propulsive phases of the countermovement jump.

Additional variables that were used to complement the core measures included the eccentric phase duration (dt_ecc), propulsion phase duration (dt_PP), and minimum vertical displacement (h_Z_min). Eccentric phase duration was defined as the time from movement onset (5% of h_Z_min) to the lowest vertical displacement point, whereas propulsion phase duration captured the time from h_Z_min to take-off. Minimum vertical displacement represented the lowest vertical position of the hip marker during the eccentric phase (i.e., depth of the countermovement), indirectly reflecting the jump technique and strategy [5,6,7,22]. 

All variables were derived for each marker (shoulder, hip, and knee), both from the 3D motion capture system (Qualisys) and the 2D video analysis (Kinovea). Measurement landmarks and phase definitions were standardized across both measurement systems to ensure methodological consistency and facilitate direct comparison between the 2D and 3D approaches.

Vertical displacement trajectories were analyzed to identify the key phases of the countermovement jump (CMJ) based on the displacement–time (Z–t) signal. The minimum vertical displacement (h_Z_min) was first identified as the lowest point of the signal, representing the transition between the eccentric (downward) and concentric (upward) phases. The onset of movement (start of the eccentric phase) was then defined as the frame where vertical displacement decreased by 5% of the total downward displacement relative to h_Z_min, marking the beginning of the countermovement.

Similarly, take-off (end of the concentric phase) was determined as the point where the displacement increased by 5% of the total upward displacement from h_Z_min, corresponding to the loss of ground contact. The duration of the eccentric phase (dt_ecc) was then determined as the time from movement onset until h_Z_min, while the duration of the concentric phase (dt_PP) was the time from h_Z_min until take-off, and the total take-off time (dt_take_off) was the total time from movement onset until take-off. The highest point of the trajectory (h_Z_max) represented the maximum vertical displacement.

### 2.5. Statistical Analysis

Descriptive statistics (Mean ± SD) were calculated for all CMJ variables obtained from both raters using the Kinovea software and the Qualisys marker-based motion capture (MoCap) system. To evaluate inter-rater reliability, intraclass correlation coefficients (ICCs (3,1)) with 95% confidence intervals (95% CIs), coefficients of variation (CV%), and standard error of measurement (SEM) were calculated to quantify relative and absolute reliability indices. In addition to that, Bland–Altman analysis was performed to assess systematic bias and agreement between raters [23].

To examine the validity of the 2D video analysis relative to the 3D motion capture reference, ICC with 95% CI, CV%, SEM, and mean differences (MDs) between the two measurement systems were computed. Additionally, Pearson’s correlation coefficients (r) were calculated to assess the strength of the linear association between methods. Correlation magnitudes were interpreted according to Hinkle et al. [24], with thresholds defined as: r < 0.30 (negligible), r = 0.30–0.50 (low), r = 0.50–0.70 (moderate), r = 0.70–0.90 (high), and r = 0.90–1.00 (very high). Furthermore, Bland–Altman analysis [23] was performed to assess systematic bias and upper/lower limits of agreement (±1.96 * SD) between raters compared to the “gold standard” system. All reliability and validity analyses were done separately for each anatomical landmark (shoulder, hip, knee, and CoM). The level of statistical significance was set at *p* < 0.05. All statistical analyses were performed using SPSS (Version 25.0, IBM Corp., Armonk, NY, USA). Bland–Altman analyses and Pearson’s r calculations were conducted using a custom MATLAB toolbox [25]. 

## 3. Results

Descriptive statistics (Mean ± SD) for core CMJ variables (dt_ecc and h_Z_max), as measured by both raters in the Kinovea (i.e., less experienced and experienced) and Qualisys systems across the shoulder, hip, and knee markers, are presented in Figure 4.

The results for the complementary CMJ variables (dt_ecc, dt_PP, and h_Z_min) are presented in Table 2.

### 3.1. Inter-Rater Reliability Results

Table 3 presents the consistency of measurements obtained by both raters using Kinovea.

### 3.2. Validity Results

The level of agreement between raters and Qualisys is illustrated in the ICC plots with 95% CI (Figure 5). Complementary validity results (CV%, SEM, r) are presented in Table 4 for both the core and complementary CMJ variables.

Bland–Altman bias and 95% limits of agreement results for core CMJ variables are presented in Figure 6 and Figure 7 for both raters compared to the “golden standard”.

Regarding complementary CMJ variables (dt_ecc, dt_PP, and h_Z_min), when comparing the less experienced rater to Qualisys, the Bland–Altman analysis revealed slightly greater variability across markers. At the shoulder marker, the mean bias was −0.017 s for dt_ecc (LoA: −0.126 to +0.092 s) and +0.004 s for dt_PP (LoA: −0.029 to +0.037 s), as well as +0.022 m for h_Z_min (LoA: −0.095 to +0.139 m). The hip marker yielded a mean bias of +0.070 s for dt_ecc (LoA: −0.036 to +0.177 s), +0.0003 s for dt_PP (LoA: −0.056 to +0.057 s), and +0.039 m for h_Z_min (LoA: −0.052 to +0.130 m). While the knee marker showed mean biases of −0.021 s for dt_ecc (LoA: −0.106 to +0.064 s), −0.006 s for dt_PP (LoA: −0.075 to +0.063 s), and +0.015 m for h_Z_min (LoA: −0.016 to +0.046 m).

Similar results were observed when comparing the experienced rater to Qualisys. At the shoulder marker, the mean bias was −0.023 s for dt_ecc (LoA: −0.119 to +0.074 s), +0.002 s for dt_PP (LoA: −0.037 to +0.041 s), and +0.039 m for h_Z_min (LoA: −0.097 to +0.175 m). At the hip marker, the bias was +0.070 s for dt_ecc (LoA: −0.029 to +0.168 s), −0.002 s for dt_PP (LoA: −0.068 to +0.065 s), and +0.033 m for h_Z_min (LoA: −0.063 to +0.128 m). Finally, the knee marker exhibited mean biases of −0.007 s for dt_ecc (LoA: −0.133 to +0.119 s), −0.001 s for dt_PP (LoA: −0.073 to +0.069 s), and +0.006 m for h_Z_min (LoA: −0.014 to +0.026 m).

## 4. Discussion

The purpose of this study was to evaluate the feasibility of a semi-automated 2D video analysis approach for assessing CMJ performance using Kinovea’s linear kinematics tracking tool. Specifically, the study aimed to (1) examine inter-rater reliability between raters of different levels of experience with the software (i.e., less experienced and experienced raters) and (2) determine the concurrent validity of this semi-automated 2D method against a “gold standard” 3D marker-based motion capture system (Qualisys).

Overall, the main findings demonstrated high inter-rater reliability and strong concurrent validity of the Kinovea-based analysis for core CMJ variables across all markers, with minimal differences between raters of different experience levels. These results indicate that the semi-automated 2D approach can provide accurate and reproducible CMJ assessments comparable to 3D motion capture systems, supporting its practical use in both research and applied sport settings.

### 4.1. Descriptive Results

Descriptive results demonstrated that mean values obtained from Kinovea were closely aligned with those recorded by the 3D reference system, regardless of rater experience. This indicates that the semi-automated tracking feature can produce consistent and comparable CMJ measurements in both experienced and less experienced users. These findings are consistent with prior validation studies of low-cost, markerless motion-tracking systems, which reported highly reproducible Kinovea-derived kinematic measurements relative to “gold standard” [3,14]. Across all markers, both raters exhibited a tendency to underestimate temporal variables and to overestimate vertical displacement relative to Qualisys, indicating a systematic difference rather than random variability in measurements. Such bias is consistent with prior findings where 2D video analyses slightly overestimate displacement due to perspective error and frame interpolation [26,27]. Importantly, these differences were small in magnitude and within practically acceptable limits [28], suggesting that Kinovea maintains adequate measurement accuracy for applied use. The absolute CMJ performance values obtained in this study reflect typical outcomes for recreationally active male basketball players, with average maximum vertical displacement between 0.45–0.48 m at the shoulder and hip, and correspondingly lower at the knee [7,21]. Temporal durations were also consistent with previously published benchmarks, confirming the validity of the testing procedure and suggesting that participants performed the CMJ with technically sound and reproducible movement patterns [5].

### 4.2. Inter-Rater Reliability

Overall, inter-rater reliability across all markers was good to excellent, with ICCs ranging from 0.73 to 0.99, CV% < 10%, and low SEM and mean differences for all variables. These findings indicate strong consistency between raters of different experience levels and support the robustness of Kinovea’s semi-automated tracking algorithm.

The hip marker produced the strongest inter-rater agreement, with ICC values ≥ 0.95 for most variables (dt_take_off, h_Z_max), CV% < 2%, and minimal absolute errors. Large, stable anatomical landmarks such as the greater trochanter allow the algorithm to perform with high precision and minimal influence from user experience [22,29]. The knee marker also demonstrated high inter-rater agreement (ICC = 0.87–0.97), particularly for the core CMJ variables (dt_take_off and h_Z_max), where differences between raters remained below 0.02 s and 0.02 m. Slightly lower reliability for dt_PP (ICC ≈ 0.87) suggests that small discrepancies in marker placement or local segment motion can propagate through the trajectory and influence temporal variable estimation. The shoulder marker showed the lowest agreement (ICC = 0.73–0.93), particularly for h_Z_max (CV ≈ 9.6%). This reduction is consistent with the biomechanical nature of the countermovement, where trunk inclination and greater angular motion can lead to temporary occlusions and small losses in marker visibility.

Despite these challenges, even the shoulder data demonstrated acceptable reproducibility for most variables. In the context of this study, this suggests that the semi-automated approach may reduce rater-dependent error compared with traditional manual 2D analysis methods [16,26], while some degree of rater influence may still remain. Bland–Altman analyses confirmed these trends, demonstrating narrow mean bias and 95% limits of agreement (within ±0.02 s for temporal and ±0.02 m for displacement variables). These results correspond well with prior work showing that differences of this magnitude are practically trivial and within acceptable error thresholds for kinematic assessments [23,28].

### 4.3. Validity Results

#### 4.3.1. Less Experienced Rater vs. Qualisys

When compared to Qualisys, the less experienced rater achieved good-to-excellent validity across most markers and variables (ICC = 0.68–0.95, CV% < 8%, low SEM, moderate-to-high r). The hip marker demonstrated the strongest agreement for h_Z_max (ICC = 0.95, r = 0.90), likely due to its proximity to the body’s center of mass and relatively stable movement trajectory during the jump [5]. However, the agreement for dt_take_off (ICC ≈ 0.75, r = 0.65) was somewhat lower than for other markers. This reduction is consistent with lower agreement observed for dt_ecc (ICC = 0.68; r = 0.62), likely reflecting brief occlusions or tracking interruptions during the descent into the squat phase. These discrepancies are more likely attributable to the increased movement dynamics and momentary loss of marker visibility in the eccentric phase rather than to rater-related factors [29].

Conversely, the knee marker demonstrated the highest level of agreement with the Qualisys system for temporal variables (ICC > 0.90, r > 0.85, CV% < 5%, SEM < 0.02 s), indicating that the knee trajectory provides a stable reference for event detection. The knee also showed high feasibility for estimating vertical displacement (ICC > 0.92, r > 0.87, CV% < 8%, SEM < 0.02 m), supporting its use for derived variables such as the reactive strength index-modified (RSImod). The shoulder marker produced the most variable outcomes for both temporal and displacement measures (ICC = 0.68–0.87, r = 0.48–0.82), which can be attributed to larger signal noise associated with trunk motion and changes in body inclination during the countermovement. Nevertheless, Bland–Altman analyses revealed only small systematic biases (≤0.02 s; ≤0.04 m), indicating that the absolute differences were negligible in practical terms. Collectively, these findings indicate that, in the context of the present study, a rater with limited prior experience was able to obtain valid results for key CMJ parameters using Kinovea, provided that recording quality and marker visibility were adequate. This suggests that the semi-automated nature of the tracking tool may reduce the influence of user expertise under controlled conditions, without implying complete independence from rater experience.

#### 4.3.2. Experienced Rater vs. Qualisys

Similarly, the experienced rater achieved strong agreement across all markers and variables, with ICC values ranging from 0.78 to 0.99, r > 0.80, CV% < 8%, and SEM < 0.03 s for temporal and <0.03 m for displacement variables.

The knee again demonstrated the highest validity, particularly for the temporal variables, where ICCs exceeded 0.90, and CV% remained below 5%, confirming its robustness for detecting jump events. Both the shoulder and hip markers demonstrated high validity for vertical displacement, particularly for h_Z_max (ICC > 0.93, r > 0.8, CV% < 4%, SEM < 0.02 m). These results are comparable to previous findings validating smartphone-based and video-tracking methods for vertical jump assessment [14,15,26]. However, lower agreement was observed for dt_ecc and dt_take_off (ICC = 0.68–0.82, CV% < 9%). These differences fall within ranges that have previously been reported as acceptable in comparable kinematic reliability studies [16,27]. Short periods of marker occlusion or rapid movement transitions may explain these discrepancies.

Taken together, both raters achieved comparable validity levels relative to the Qualisys system. Rather than indicating that user experience has a negligible role in measurement accuracy, these results suggest that factors such as video quality, marker visibility, and consistency of jump execution play a primary role in determining measurement quality when using Kinovea’s semi-automated features. This interpretation is consistent with findings from recent validation studies [5,22].

### 4.4. Study Limitations

Several methodological limitations should be acknowledged when interpreting the present findings, which may guide improvements in future applications. First, occasional marker occlusions were observed during the recordings, particularly when participants wore clothing with lower visual contrast relative to the marker color (e.g., light shorts with light markers) or when excessive trunk inclination reduced visibility of the hip or shoulder marker in the sagittal plane. Such interruptions likely introduced small tracking noise, especially in the eccentric phase, consistent with prior observations in both marker-based and markerless motion capture [3,29]. To minimize these effects, it is essential to use high-contrast marker–clothing combinations and instruct participants to avoid excessive forward leaning when performing the jump [21].

Second, the 2D analysis is inherently constrained by its single-plane perspective, which increases sensitivity to camera alignment and calibration. Accurate calibration is essential, as the scaling factor determines all subsequent displacement values when using Kinovea’s linear kinematics tracking tool. Small deviations in the calibration of height can propagate through the displacement–time curve and systematically affect the estimation of key variables (Figure 8).

As illustrated in Figure 8, even slight errors in calibration may lead to overestimation or underestimation of vertical displacement, which likely contributed to the modest upward bias in h_Z_max observed in this study. Because temporal variables such as dt_take_off and dt_PP are derived from changes in the displacement signal, minor calibration errors can also shift the timing of movement events, particularly around the minimum displacement and take-off points. These considerations highlight the need for consistent and carefully executed calibration procedures, as calibration accuracy represents one of the most influential user-dependent factors in 2D kinematic analysis.

In this study, 2D calibration in Kinovea was performed using participants’ body height rather than an external rigid calibration object. This approach was chosen as a pragmatic solution to facilitate data collection in real-world environments; however, it introduces an important methodological limitation. Small deviations in posture during calibration, such as slight knee flexion, incomplete upright alignment, or minor head movement, can systematically shift the entire vertical displacement curve and lead to overestimation or underestimation of signal amplitude. It is important to emphasize that this systematic bias is not inherent to the Kinovea software itself but rather arises from the stature-based calibration method. To minimize these errors, future studies should use an external object of known height, placed in the sagittal plane and aligned vertically, as suggested in previous 2D validation studies [16,30].

While the title reflects the potential transition from laboratory to field applications, all measurements in the present study were obtained under controlled laboratory conditions. Although the calibration procedure was intentionally chosen to reflect field-based practices, the experimental setup still relied on optimized recording conditions and placement of reflective markers. These conditions do not fully capture the variability typically encountered in real-world field environments such as sports halls or training facilities. Nevertheless, the minimal hardware requirements of Kinovea, which relies solely on standard video recordings rather than specialized motion capture infrastructure, support its strong potential for field-based applications and position it as a practical bridge between laboratory-grade motion analysis and everyday sport performance monitoring. Consequently, the validity and reliability of the Kinovea approach under more demanding field conditions should be confirmed in future studies.

Although the semi-automated tracking algorithm reduces user-related variability, overall measurement accuracy still depends on recording conditions such as camera resolution, frame rate, lighting, and calibration stability. High-quality, well-lit recordings remain essential for extracting reliable trajectories, a requirement similarly emphasized in other video-based and markerless motion capture systems [3,15,20]. A further limitation involves the use of reflective markers placed directly on anatomical landmarks, which does not fully replicate performance in markerless applications. Future work should examine how accurately Kinovea’s linear kinematics tool performs when no physical markers are present, and trajectories are estimated solely from visible anatomical features, an area gaining increasing attention in emerging AI-based motion capture technologies [20,31].

Finally, the sample used in this study consisted exclusively of recreationally active male basketball players who were already familiar with the CMJ technique and demonstrated consistent jump execution. While the statistical power analysis confirmed that nine participants were sufficient to meet the methodological aims of the study, the homogeneity of the sample introduces an important limitation. Therefore, extending these findings to other populations, such as athletes from different sports, female athletes, older adults, or clinical cohorts, requires careful consideration.

This is particularly relevant given that movement variability is a major contributor to kinematic error [6]. Populations with lower technical proficiency or tasks with greater movement complexity may exhibit increased variability, potentially influencing both displacement trajectories and the accuracy of semi-automated tracking. Future research should therefore evaluate performance across a broader and more diverse range of participants and movement patterns to determine whether the high levels of reliability and concurrent validity observed here extend beyond trained male basketball players.

Despite these limitations, the results confirm that the semi-automated analysis of CMJ performance in Kinovea provides a feasible and practical alternative for analyzing vertical jump performance. Given its accessibility and simplicity, it represents a valuable bridge between laboratory-grade motion analysis and applied sports performance monitoring [22,26].

## 5. Conclusions

This study demonstrated that Kinovea’s semi-automated 2D linear kinematics tool provides valid and reliable estimates of key countermovement jump (CMJ) variables when compared to a 3D marker-based motion capture system. Across both raters, the knee marker proved to be the most accurate and consistent for detecting temporal events such as the time to take-off, while the hip marker demonstrated the highest validity for estimating maximum vertical displacement. These results indicate that Kinovea’s linear kinematics tool can be effectively used for assessing CMJ performance, provided that marker visibility and video quality are adequately controlled.

Importantly, in the context of the present study, the less experienced rater obtained results comparable to those of the experienced rater, suggesting that the semi-automated nature of the analysis may reduce the influence of user experience under controlled conditions. Instead, factors such as calibration accuracy, video resolution, lighting conditions, contrast between the subject’s clothing and markers, and consistency of jump execution appeared to play a more decisive role in ensuring data quality.

From a practical perspective, these results support the use of Kinovea as a cost-effective and accessible alternative for CMJ analysis in both research and applied sport settings. However, it is important to note that reliable CMJ assessment using Kinovea depends on maintaining adequately controlled recording conditions. Specifically, a fixed camera should be positioned perpendicular to the sagittal plane, with sufficient lighting and clear visual contrast between the participant’s clothes/shoes and the background, and recordings should be acquired at an appropriately high frame rate (i.e., 120 or 240 fps). Furthermore, calibration procedures should be executed with care, and the use of an external object of known dimensions is recommended whenever feasible to minimize scaling error. Marker placement, or the selection of clearly visible anatomical landmarks, should aim to reduce occlusion, particularly during the eccentric phase of the movement. When recording conditions are properly optimized, this semi-automated 2D approach can deliver results comparable to laboratory-grade systems, offering a feasible solution for routine athlete monitoring, field testing, and applied performance assessments.

## Figures and Tables

**Figure 1 sensors-26-00024-f001:**
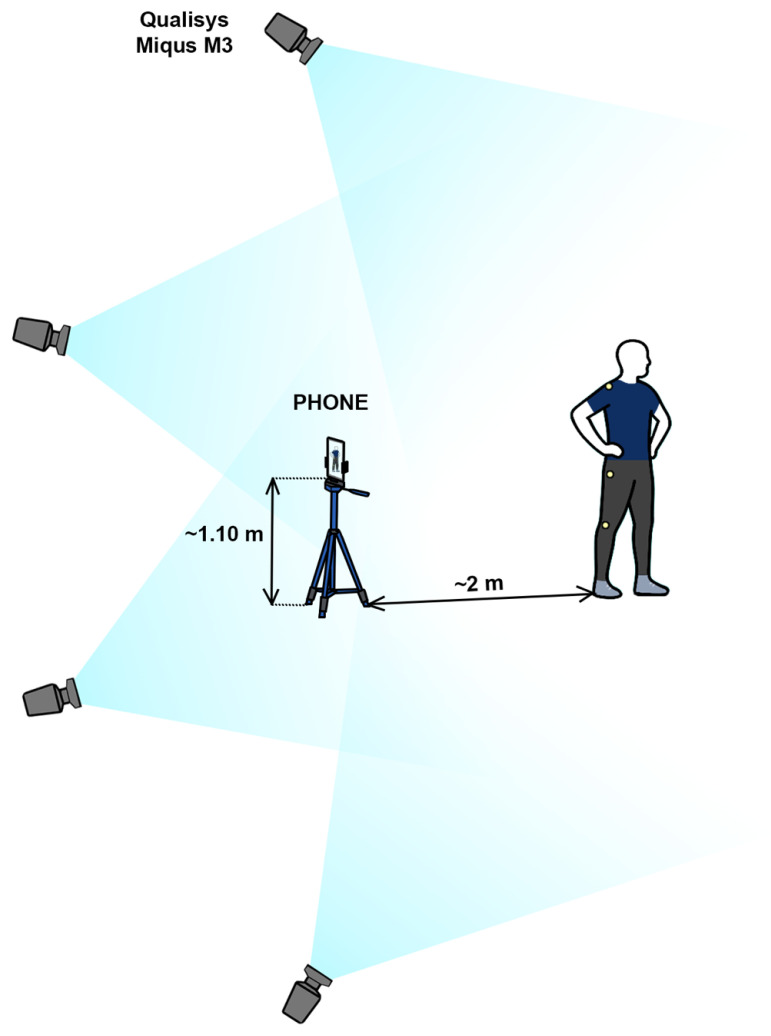
Illustration of the measurement setup.

**Figure 2 sensors-26-00024-f002:**
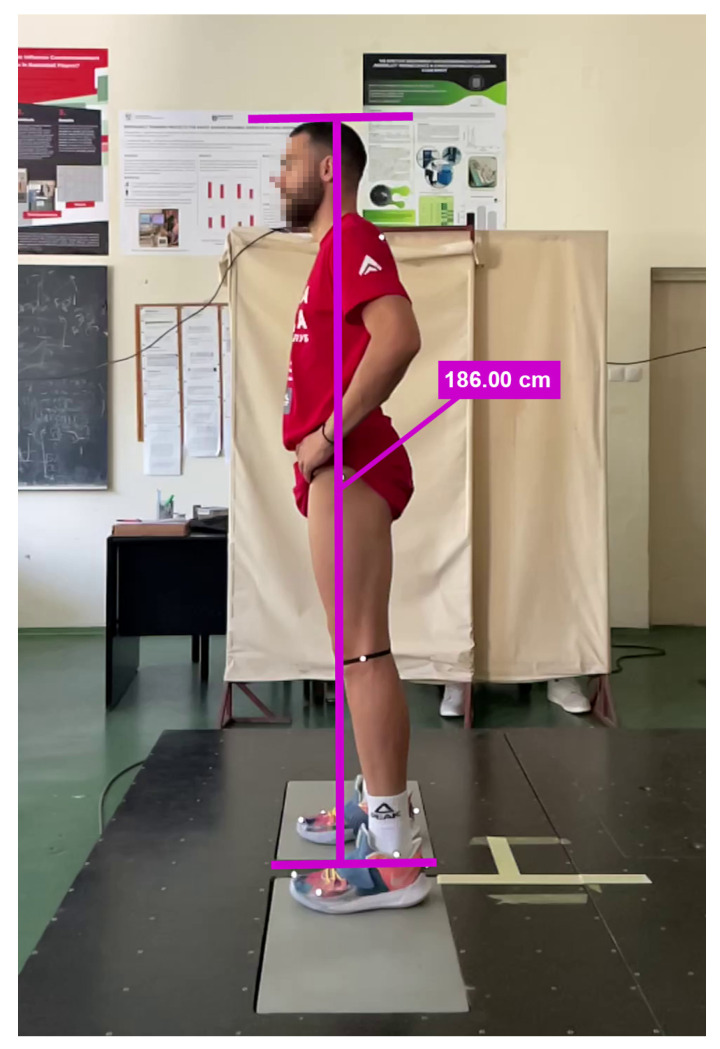
Calibration process in Kinovea based on body height.

**Figure 3 sensors-26-00024-f003:**
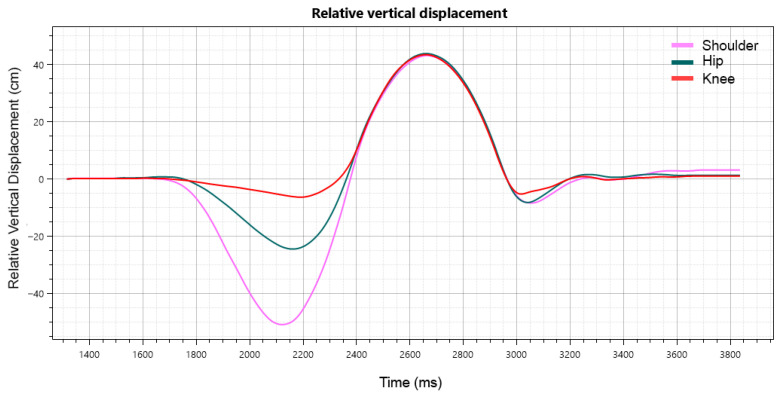
Vertical displacement trajectories derived from Kinovea.

**Figure 4 sensors-26-00024-f004:**
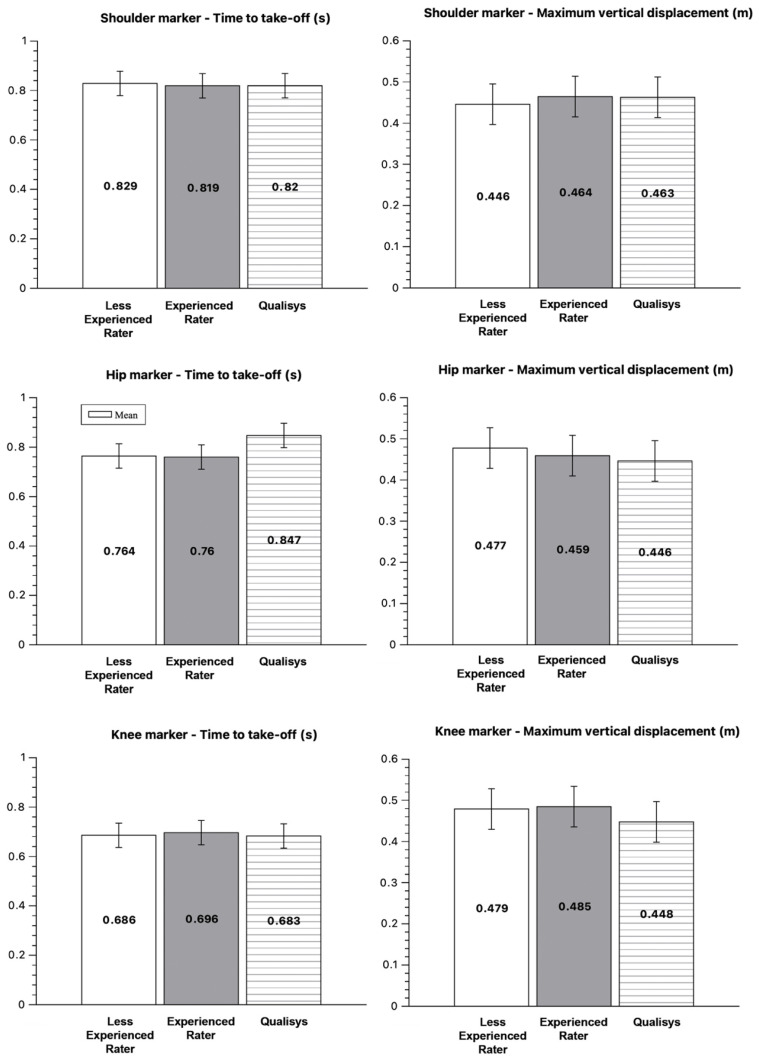
Descriptive statistics (mean ± SD) for time to take-off (dt_take_off, left panels) and maximum vertical displacement (h_Z_max, right panels) obtained from Kinovea by the less experienced and experienced raters and from the Qualisys system. Results are shown separately for the shoulder (**top row**), hip (**middle row**), and knee (**bottom row**) markers. Error bars represent standard deviations.

**Figure 5 sensors-26-00024-f005:**
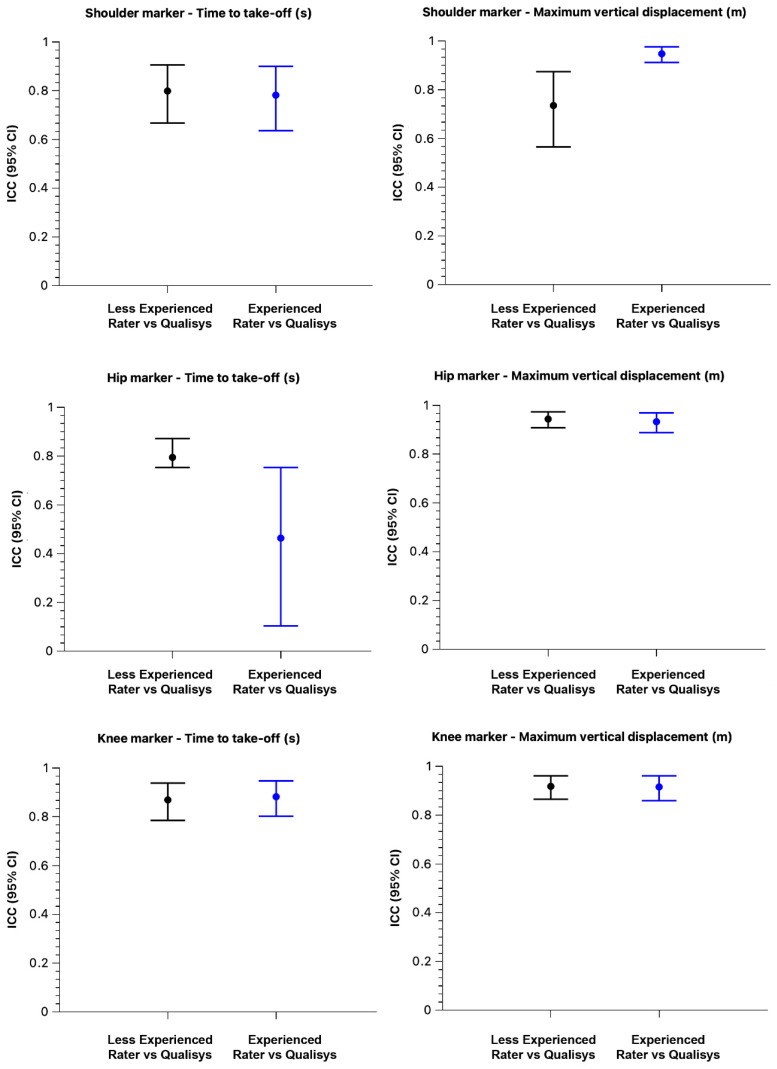
ICC plots with 95% confidence intervals for core CMJ variables.

**Figure 6 sensors-26-00024-f006:**
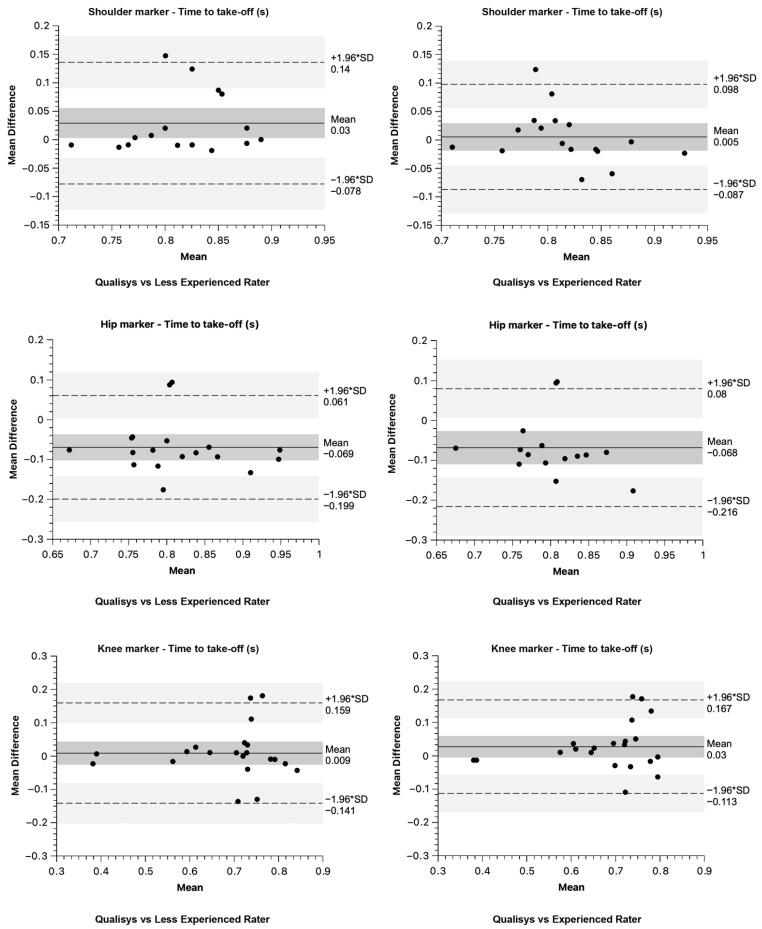
Bland–Altman plots for Time to take-off difference between raters and Qualisys.

**Figure 7 sensors-26-00024-f007:**
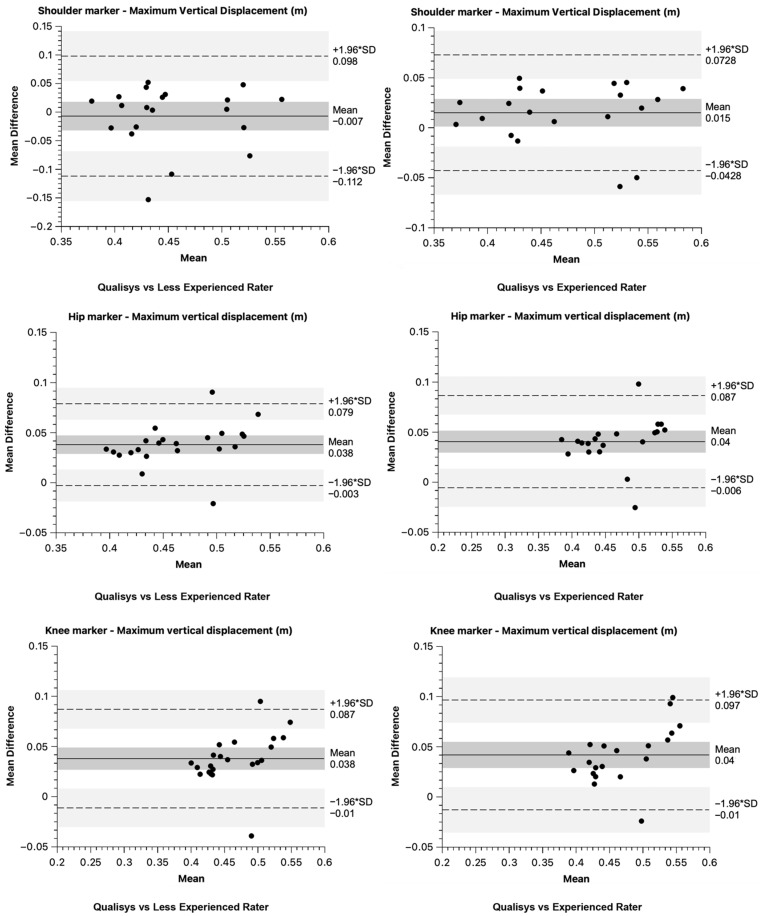
Bland–Altman plots for the maximum vertical displacement difference between raters and Qualisys.

**Figure 8 sensors-26-00024-f008:**
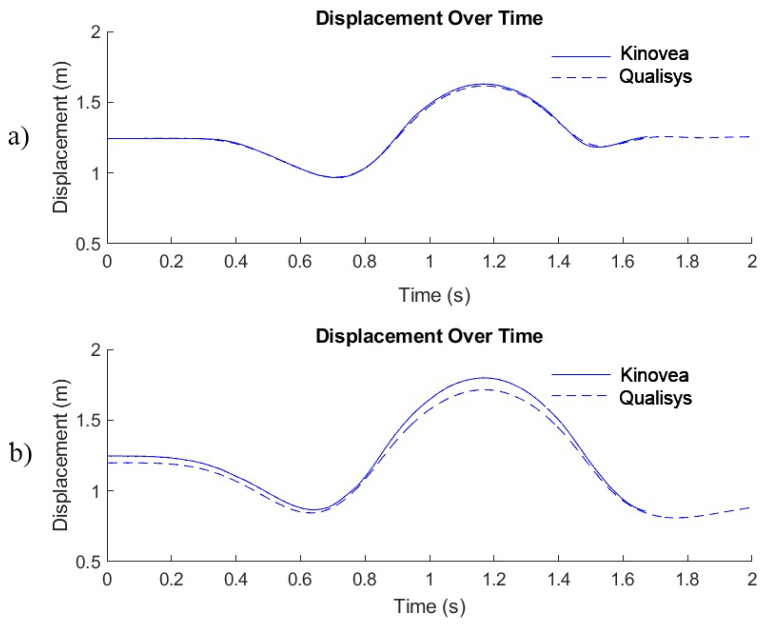
Vertical displacement–time (Z–t) curves for the countermovement jump (CMJ) obtained from Qualisys (solid line) and Kinovea (dashed line): (**a**) correct height calibration in Kinovea; (**b**) incorrect height calibration in Kinovea, resulting from slight misalignment of the body-height reference segment in Kinovea. This produced a systematic vertical offset and inflated displacement amplitude when compared with the simultaneously recorded Qualisys curve.

**Table 1 sensors-26-00024-t001:** Overview of kinematic variables in this study.

Variable	Explanation	Unit
dt_ecc	Duration of eccentric phase	s
dt_PP	Duration of propulsion phase	s
dt_take_off	Total time until take-off	s
h_Z_max	Maximum vertical displacement	m
h_Z_min	Minimum vertical displacement	m

Note: s = seconds; m = meters; m/s = meters per second.

**Table 2 sensors-26-00024-t002:** Descriptive results for the complementary CMJ variables.

		Kinovea				Qualisys	
		Less Experienced Rater		Experienced Rater			
Marker	Variable	Mean	SD	Mean	SD	Mean	SD
SHOULDER	dt_ecc (s)	0.481	0.045	0.495	0.064	0.487	0.058
	dt_PP (s)	0.337	0.029	0.339	0.037	0.340	0.027
	h_Z_min (m)	−0.572	0.087	−0.590	0.086	−0.548	0.079
HIP	dt_ecc (s)	0.451	0.031	0.451	0.037	0.529	0.063
	dt_PP (s)	0.311	0.038	0.313	0.048	0.314	0.040
	h_Z_min (m)	−0.296	0.084	−0.296	0.084	−0.268	0.065
KNEE	dt_ecc (s)	0.409	0.083	0.406	0.085	0.400	0.087
	dt_PP (s)	0.288	0.044	0.280	0.051	0.283	0.038
	h_Z_min (m)	−0.100	0.035	−0.093	0.031	−0.090	0.030

Note: SD—standard deviation.

**Table 3 sensors-26-00024-t003:** Inter-rater reliability statistics for Kinovea measurements obtained by raters.

		Kinovea—Less Experienced vs. Experienced Rater			
Marker	Variable	ICC (95% CI)	CV%	SEM	MD
SHOULDER	dt_ecc (s)	0.922 (0.845 ÷ 0.961)	3.86	0.019	−0.002
	dt_PP (s)	0.875 (0.762 ÷ 0.933)	4.17	0.014	−0.002
	dt_take_off (s)	0.933 (0.876 ÷ 0.964)	3.11	0.026	−0.007
	h_Z_max (m)	0.734 (0.510 ÷ 0.855)	9.65	0.044	0.027
	h_Z_min (m)	0.984 (0.971 ÷ 0.991)	2.56	0.015	0.028
HIP	dt_ecc (s)	0.959 (0.925 ÷ 0.977)	1.66	0.008	0.001
	dt_PP (s)	0.946 (0.899 ÷ 0.971)	4.44	0.014	−0.001
	dt_take_off (s)	0.972 (0.949 ÷ 0.985)	1.91	0.015	0.002
	h_Z_max (m)	0.989 (0.979 ÷ 0.994)	1.15	0.006	0.002
	h_Z_min (m)	0.995 (0.992 ÷ 0.998)	1.84	0.005	−0.003
KNEE	dt_ecc (s)	0.906 (0.909 ÷ 0.973)	5.97	0.024	0.003
	dt_PP (s)	0.867 (0.769 ÷ 0.926)	6.11	0.017	0.009
	dt_take_off (s)	0.965 (0.937 ÷ 0.981)	3.1	0.021	0.012
	h_Z_max (m)	0.904 (0.831 ÷ 0.947)	3.53	0.017	0.001
	h_Z_min (m)	0.910 (0.840 ÷ 0.950)	10.07	0.01	−0.008

Note: ICC—intra-class correlation coefficient; 95% CI—95% confidence Interval; SEM—standard error of measurement; MD—mean difference.

**Table 4 sensors-26-00024-t004:** Validity statistics for Kinovea (less experienced and experienced rater) vs. Qualisys measurements.

		Kinovea (Less Experienced Rater) vs. Qualisys				
Marker	Variable	ICC (95% CI)	CV%	SEM	MD	r
SHOULDER	dt_ecc (s)	0.678 (0.400 ÷ 0.827)	8.06	0.039	0.023	0.481
	dt_PP (s)	0.868 (0.754 ÷ 0.929)	4.04	0.014	−0.002	0.818 ***
	dt_take_off (s)	0.823 (0.667 ÷ 0.906)	4.81	0.039	0.029	0.718 ***
	h_Z_max (m)	0.766 (0.565 ÷ 0.874)	8.37	0.038	−0.007	0.563 **
	h_Z_min (m)	0.785 (0.600 ÷ 0.884)	8.44	0.048	−0.039	0.687 ***
HIP	dt_ecc (s)	0.678 (0.394 ÷ 0.829)	7.11	0.035	−0.069	0.624 **
	dt_PP (s)	0.835 (0.692 ÷ 0.911)	7.35	0.023	0.002	0.756 ***
	dt_take_off (s)	0.758 (0.535 ÷ 0.872)	5.7	0.046	−0.069	0.648 **
	h_Z_max (m)	0.950 (0.908 ÷ 0.973)	3.21	0.015	0.038	0.902 ***
	h_Z_min (m)	0.882 (0.780 ÷ 0.937)	12.12	0.034	−0.033	0.819 ***
KNEE	dt_ecc (s)	0.837 (0.697 ÷ 0.913)	10.89	0.044	0.007	0.752 ***
	dt_PP (s)	0.816 (0.657 ÷ 0.901)	8.78	0.025	0.001	0.743 ***
	dt_take_off (s)	0.884 (0.785 ÷ 0.938)	7.69	0.053	0.009	0.820 ***
	h_Z_max (m)	0.927 (0.865 ÷ 0.961)	3.84	0.018	0.038	0.875 ***
	h_Z_min (m)	0.971 (0.946 ÷ 0.984)	7.96	0.007	−0.006	0.943 ***
		**Kinovea (Experienced Rater) vs. Qualisys**				
SHOULDER	dt_ecc (s)	0.500 (0.035 ÷ 0.741)	8.86	0.043	0.017	0.251
	dt_PP (s)	0.901 (0.811 ÷ 0.948)	3.51	0.012	−0.004	0.815 ***
	dt_take_off (s)	0.809 (0.636 ÷ 0.900)	4.03	0.033	0.005	0.653 **
	h_Z_max (m)	0.954 (0.912 ÷ 0.976)	4.54	0.021	0.015	0.902 ***
	h_Z_min (m)	0.850 (0.715 ÷ 0.921)	7.55	0.042	−0.022	0.738 ***
HIP	dt_ecc (s)	0.478 (−0.007 ÷ 0.729)	8.31	0.041	−0.070	0.316
	dt_PP (s)	0.850 (0.710 ÷ 0.922)	6.36	0.019	0.000	0.743 ***
	dt_take_off (s)	0.530 (0.104 ÷ 0.753)	6.77	0.054	−0.068	0.366
	h_Z_max (m)	0.941 (0.888 ÷ 0.969)	3.6	0.017	0.041	0.902 ***
	h_Z_min (m)	0.887 (0.784 ÷ 0.941)	−11.64	0.033	−0.039	0.853 ***
KNEE	dt_ecc (s)	0.929 (0.862 ÷ 0.963)	7.62	0.031	0.021	0.872 ***
	dt_PP (s)	0.782 (0.584 ÷ 0.885)	8.44	0.024	0.006	0.687 ***
	dt_take_off (s)	0.897 (0.802 ÷ 0.947)	7.33	0.051	0.027	0.834 ***
	h_Z_max (m)	0.926 (0.859 ÷ 0.961)	4.22	0.02	0.042	0.909 ***
	h_Z_min (m)	0.930 (0.866 ÷ 0.963)	11.8	0.011	−0.015	0.901 ***

Note: ICC—intra-class correlation coefficient; 95% CI—95% confidence interval; SEM—standard error of measurement; MD—mean difference; r—Pearson’s correlation coefficient; **—*p* < 0.05; ***—*p* < 0.001.

## Data Availability

The data presented in this study are available upon request from the corresponding author.

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
