# Peer review of "Sensors2026, 26(1), 24;https://doi.org/10.3390/s26010024"

_sensors, 2025, doi:10.3390/s26010024_

Round 1

Reviewer 1 Report

Comments and Suggestions for Authors

Dear authors,

First, I wish to congratulate the authors on an excellently formulated and well-structured manuscript, supported by experiments conducted with notable rigor. I see immense value in this document for institutions or researchers wishing to conduct biomechanical studies but who often lack expensive tools or professional analysis systems. This study demonstrates that accessible tools like Kinovea, when used correctly, represent an excellent alternative.

Precisely to ensure that readers of this article can successfully implement this type of low-cost solution in their own laboratories, I present the following observations. The objective of these recommendations is to ensure that any researcher has total clarity regarding the parameters and limitations to consider in order to replicate the method successfully.

Specific Observations:

  1. Exclusion of the foot markers: Regarding the exclusion of the foot markers due to occlusions and signal noise, I suggest expanding on the information presented. It would be very valuable to indicate—if the data is available—what percentage of trials or frames presented occlusion issues. I also recommend adding an explicit clarification in the text stating that removing this marker was a methodological decision to ensure signal integrity, thereby avoiding excessive interpolation that could have introduced bias or artifacts. This is particularly relevant given that the study focuses on global CMJ displacement rather than specific ankle dynamics.
  2. Sample size and composition: Although the statistical power calculation justifies that 9 participants were sufficient to meet the methodological objective, the final sample (n=10) composed exclusively of males and basketball players limits the generalization of the findings. I suggest reinforcing the Limitations section, explicitly indicating that the results are representative of this demographic profile and that extrapolation to other populations (females, older adults, clinical populations, or other sports) must be done with caution.
  3. Stature-based 2D calibration: The calibration protocol utilized the participant's stature instead of an external rigid reference object. Although the authors acknowledge the implications of this, it is necessary to emphasize that small errors (such as the subject not being perfectly upright or having slight knee flexion) can systematically shift the entire vertical curve. This systematic bias is not inherent to the Kinovea software, but to the chosen calibration method. I request that you clarify in the text that this was a pragmatic decision and explicitly note that the use of external calibration patterns (objects of known dimensions) is recommended in future studies to minimize this error.
  4. Interpolation of missing data (in addition to the foot marker): The use of interpolation (linear/polynomial) to reconstruct data gaps is mentioned. While this is standard practice, the manuscript lacks quantification of this procedure. I request that you briefly indicate the approximate percentage of trajectories where interpolation was necessary and specify whether this was applied only to short gaps (a few frames) or if there were cases of more prolonged signal loss, estimating the potential level of distortion introduced in such cases.
  5. Moderation of language and scope of conclusions: There are two aspects where the statements appear overly categorical and should be nuanced: (1) Influence of experience: The conclusion that user experience has minimal influence on precision is derived from the comparison between a single novice rater and a single expert rater (n=2). This basis is insufficient to make such an absolute generalization. It would be more prudent to rephrase it as: "In the context of this study, the rater with little experience obtained results comparable to the experienced rater, suggesting that the semi-automated nature of the tool may reduce the influence of user expertise. (2) "Generalization "From Laboratory to Field": Although the title suggests a transition to the field, the authors must acknowledge that the data were obtained in a controlled laboratory environment (uniform floor, optimized lighting, reflective markers, and aligned cameras). This contrasts with real field conditions (gyms, variable lighting, diverse clothing, absence of markers), where Kinovea's performance might be affected. I suggest adding an explicit phrase in the limitations: "Our results were obtained in a controlled laboratory environment; therefore, validity and reliability under more demanding field conditions must be confirmed in future studies.

Conclusion: I believe that once these corrections are introduced, the article will consolidate itself as an excellent tool and a fundamental guide for those seeking to implement low-cost biomechanical analysis systems. Due to the high practical value of the work, I have recommended to the Editor that the article be accepted subject to these minor modifications.

Comments on the Quality of English Language

The manuscript is written in clear and professional English.

Author Response

Comment 1: Exclusion of the foot markers: Regarding the exclusion of the foot markers due to occlusions and signal noise, I suggest expanding on the information presented. It would be very valuable to indicate—if the data is available—what percentage of trials or frames presented occlusion issues. I also recommend adding an explicit clarification in the text stating that removing this marker was a methodological decision to ensure signal integrity, thereby avoiding excessive interpolation that could have introduced bias or artifacts. This is particularly relevant given that the study focuses on global CMJ displacement rather than specific ankle dynamics.

Answer: Thank you for this constructive suggestion. We have expanded the Methods section to provide a clearer and more detailed rationale for the exclusion of the foot (toe) marker. We now explicitly state that the marker exhibited frequent occlusions and increased signal noise across multiple trials, which would have required extensive interpolation across multiple frames. To preserve signal integrity and minimize the risk of bias associated with excessive interpolation, the toe marker was therefore excluded as a deliberate methodological decision. We further clarify that this choice is appropriate given that the primary aim of the study was to assess global CMJ displacement and temporal characteristics rather than ankle- or foot-specific kinematics. These clarifications have been incorporated into the revised manuscript (page 4, lines 141-154).

Comment 2: Sample size and composition: Although the statistical power calculation justifies that 9 participants were sufficient to meet the methodological objective, the final sample (n=10) composed exclusively of males and basketball players limits the generalization of the findings. I suggest reinforcing the Limitations section, explicitly indicating that the results are representative of this demographic profile and that extrapolation to other populations (females, older adults, clinical populations, or other sports) must be done with caution.

Answer: Thank you for highlighting the importance of addressing the sample characteristics and their impact on generalizability. We have expanded the Study Limitations section to more clearly acknowledge that the sample consisted exclusively of recreationally active male basketball players and that the findings are therefore most representative of this demographic profile. We also emphasize that generalizability to other populations should be made with caution. These revisions are now included in the manuscript (page 19, lines 530-544).

Comment 3: Stature-based 2D calibration: The calibration protocol utilized the participant's stature instead of an external rigid reference object. Although the authors acknowledge the implications of this, it is necessary to emphasize that small errors (such as the subject not being perfectly upright or having slight knee flexion) can systematically shift the entire vertical curve. This systematic bias is not inherent to the Kinovea software, but to the chosen calibration method. I request that you clarify in the text that this was a pragmatic decision and explicitly note that the use of external calibration patterns (objects of known dimensions) is recommended in future studies to minimize this error.

Answer: Thank you for this important and constructive comment. We have revised the Study Limitations section to explicitly clarify that the use of stature-based calibration was a pragmatic decision aimed at facilitating data collection in applied and real-world settings. We now emphasize that small postural deviations during calibration may systematically shift the entire vertical displacement curve and influence displacement amplitude. Importantly, we clearly state that this systematic bias is not inherent to the Kinovea software itself, but rather a consequence of the chosen calibration approach. In addition, we now explicitly recommend the use of external calibration objects of known dimensions, aligned with the sagittal plane, as a preferred strategy for future studies to minimize calibration-related error (page 18, lines 496–506).

Comment 4: Interpolation of missing data (in addition to the foot marker): The use of interpolation (linear/polynomial) to reconstruct data gaps is mentioned. While this is standard practice, the manuscript lacks quantification of this procedure. I request that you briefly indicate the approximate percentage of trajectories where interpolation was necessary and specify whether this was applied only to short gaps (a few frames) or if there were cases of more prolonged signal loss, estimating the potential level of distortion introduced in such cases.

Answer: Thank you for this valuable comment. We have revised the Methods section to clarify the scope and nature of the interpolation procedure. Specifically, we now state that interpolation was required only for brief gaps caused by momentary marker occlusions during dynamic movement phases, typically spanning only a few consecutive frames, and occurring in a small proportion of the recorded trajectories. Importantly, no cases of prolonged signal loss were observed. We further clarify that this conservative application of interpolation aligns with standard practices in marker-based motion capture analysis and is unlikely to have introduced meaningful distortion to the displacement–time signals or derived kinematic variables (page 5, lines 183–189).

Comment 5: Moderation of language and scope of conclusions: There are two aspects where the statements appear overly categorical and should be nuanced: (1) Influence of experience: The conclusion that user experience has minimal influence on precision is derived from the comparison between a single novice rater and a single expert rater (n=2). This basis is insufficient to make such an absolute generalization. It would be more prudent to rephrase it as: "In the context of this study, the rater with little experience obtained results comparable to the experienced rater, suggesting that the semi-automated nature of the tool may reduce the influence of user expertise. (2) "Generalization "From Laboratory to Field": Although the title suggests a transition to the field, the authors must acknowledge that the data were obtained in a controlled laboratory environment (uniform floor, optimized lighting, reflective markers, and aligned cameras). This contrasts with real field conditions (gyms, variable lighting, diverse clothing, absence of markers), where Kinovea's performance might be affected. I suggest adding an explicit phrase in the limitations: "Our results were obtained in a controlled laboratory environment; therefore, validity and reliability under more demanding field conditions must be confirmed in future studies.

Answer: Thank you for this thoughtful comment. We have revised the manuscript to moderate the interpretation and scope of our conclusions. First, statements regarding the influence of user experience have been rephrased in the Discussion and Conclusions sections to explicitly reflect the context of the present study, emphasizing that the less experienced rater obtained results comparable to the experienced rater under controlled conditions, and suggesting that the semi-automated nature of the tool may reduce the influence of user expertise, rather than making a generalized claim (page 16, lines 406-409, page 17, lines 437-442 and 458-463, and page 20, lines 559-564).
Second, we have expanded the Study Limitations section to explicitly acknowledge that all data were collected in a controlled laboratory environment, including standardized lighting, camera alignment, and the use of reflective markers. We now clearly state that the validity and reliability of the Kinovea approach under more demanding field conditions must be confirmed in future research (pages 18-19, lines 507-518).

Reviewer 2 Report

Comments and Suggestions for Authors

The authors compared two motion capture techniques is interesting and valuable, but I believe there are still some aspects that could be improved.

  1. Lines 117-119 mention using G*Power to calculate sample size based on a correlation of 0.8, but does not explain why this effect size was chosen.
  2.  Only 10 "recreationally active male basketball players" were included, but no clear definition of "recreationally active" is provided (e.g., weekly exercise frequency, duration).
  3.  Only male participants were included, limiting the applicability of results to female populations. This limitation should be stated.
  4.  Lines 148-150 mention rater experience differences but do not specify whether raters received standardized training. The training process, duration, and content should be described or stated.
  5. Lines 136-142 mention that the toe marker was excluded due to occlusion, but do not provide specific information about the frequency, causes, or impact of the occlusion. If the proportion of occluded markers is too high, can the motion capture results still be considered reliable?
  6. Should the subject’s face be obscured in Figure 2?
  7. Lines 242-246 mention using ICC to assess reliability but do not specify which ICC model was used (e.g., ICC(2,1) or ICC(3,1)). This is crucial for result interpretation as different models have different assumptions and interpretations. 
  8. Lines 263-265 mention Figure 4 shows descriptive statistics but do not specify what the figure presents. I am confused about this figure.
  9. Lines 487-491 state that Kinovea is a "cost-effective and accessible alternative" but do not provide specific application guidelines.
  10. The title emphasizes "from laboratory to field," but in reality, all experiments were still conducted in a laboratory setting. How can it be verified that this method is also applicable in everyday, real-world scenarios?

Author Response

Comment 1: Lines 117-119 mention using G*Power to calculate sample size based on a correlation of 0.8, but does not explain why this effect size was chosen.

Answer: Thank you for the helpful suggestion. We have clarified the rationale for the selected effect size in the Methods section. The assumed correlation coefficient of 0.8 was chosen based on previously published CMJ validation studies reporting strong associations between video-based kinematic measures and reference systems. This explanation has now been added to the manuscript (page 3, lines 121-123).

Comment 2: Only 10 "recreationally active male basketball players" were included, but no clear definition of "recreationally active" is provided (e.g., weekly exercise frequency, duration).

Answer: Thank you for this comment. We have now clarified the definition of “recreationally active” in the Methods section by specifying the minimum frequency and duration of training required for inclusion (page 3, lines 112-113).

Comment 3: Only male participants were included, limiting the applicability of results to female populations. This limitation should be stated.

Answer: Thank you for your valuable comment. We have now explicitly acknowledged this limitation in the Study Limitations section by stating that the sample consisted exclusively of recreationally active male basketball players and that extrapolation of the findings to female athletes and other populations should be made with caution. This clarification has now been added to Section 4.4 of the manuscript (page 19, lines 530-544).

Comment 4:  Lines 148-150 mention rater experience differences but do not specify whether raters received standardized training. The training process, duration, and content should be described or stated.

Answer: Thank you so much for your comment. We have clarified the rater preparation procedure in the Methods section. No formal standardized training protocol was implemented prior to analysis. Both raters were familiarized with the Kinovea linear kinematics tracking tool using the official software documentation and user guidelines (available at https://www.kinovea.org/help/en/measurement/kinematics/linear.html).
The experienced rater had prior experience using this tool, whereas the less experienced rater had no previous hands-on experience beyond this familiarization phase. This information has now been added to the manuscript (page 5, lines 162-167).

Comment 5: Lines 136-142 mention that the toe marker was excluded due to occlusion, but do not provide specific information about the frequency, causes, or impact of the occlusion. If the proportion of occluded markers is too high, can the motion capture results still be considered reliable?

Answer: Thank you for this important suggestion. We have revised the Methods section to clarify the frequency, causes, and implications of toe marker occlusion. We now explain that occlusions occurred frequently across multiple trials and were primarily caused by insufficient visual contrast between the toe marker and participants’ footwear, as most participants wore predominantly white shoes similar in color to the marker, which impaired reliable tracking in Kinovea. We further clarify that the exclusion of the toe marker does not compromise the reliability of the reported motion capture results, as the primary outcome measures were derived from proximal markers (shoulder, hip, and knee) that provided stable trajectories and were central to the assessment of global CMJ displacement and temporal variables. This methodological clarification has now been added to the manuscript.
(page 4, lines 141-154).

Comment 6: Should the subject’s face be obscured in Figure 2?

Answer: Thank you for raising this point. The subject’s face in Figure 2 has now been blurred to ensure participant anonymity. This modification has been implemented in the revised manuscript (Figure 2, page 6, line 202).

Comment 7: Lines 242-246 mention using ICC to assess reliability but do not specify which ICC model was used (e.g., ICC(2,1) or ICC(3,1)). This is crucial for result interpretation as different models have different assumptions and interpretations. 

Answer: Thank you for highlighting this important point. We have now explicitly specified the ICC model used in the Methods section. Reliability was assessed using a two-way mixed-effects model with absolute agreement for single measurements (ICC(3,1)). This clarification has been added to the manuscript (page 8, line 268).

Comment 8: Lines 263-265 mention Figure 4 shows descriptive statistics but do not specify what the figure presents. I am confused about this figure.

Answer: Thank you for pointing this out. We have revised both the manuscript text and the caption of Figure 4 to clearly specify what is presented. The figure now explicitly describes the variables, markers, and measurement systems included, and the accompanying text has been clarified to guide the reader. These revisions improve the interpretability of Figure 4 and have been incorporated into the revised manuscript (page 9, lines 297-299, and 301-305).

Comment 9: Lines 487-491 state that Kinovea is a "cost-effective and accessible alternative" but do not provide specific application guidelines.

Answer: We appreciate your valuable comment and have now expanded the Conclusions section to include practical application guidelines for the use of Kinovea in CMJ assessment. The revised text now outlines recommended recording conditions, calibration considerations, and marker or landmark visibility requirements to support reliable implementation in applied settings. These additions clarify the contexts in which Kinovea can be effectively used and have been incorporated into the revised manuscript (page 20, lines 567–575).

Comment 10: The title emphasizes "from laboratory to field," but in reality, all experiments were still conducted in a laboratory setting. How can it be verified that this method is also applicable in everyday, real-world scenarios?

Answer: Thank you for this important comment. We have revised the Study Limitations section to explicitly acknowledge that all measurements in the present study were obtained under controlled laboratory conditions and therefore do not directly confirm validity under real-world field settings. We clarify that, although the calibration procedure was intentionally selected to reflect practices feasible in field-based applications, the experimental setup relied on optimized recording conditions and reflective marker placement that may not be representative of typical field environments. At the same time, we still emphasize that the minimal hardware requirements of Kinovea, relying solely on standard video recordings rather than specialized motion capture infrastructure, support its strong potential for field-based application and position it as a practical bridge between laboratory-grade motion analysis and applied sport performance monitoring. We further state that the validity and reliability of the approach under more demanding field conditions should be confirmed in future studies. These revisions have been incorporated into the manuscript in the Study Limitations section (pages 18-19, lines 507-518). 

Round 2

Reviewer 2 Report

Comments and Suggestions for Authors

The authors have addressed my concerns and revised the manuscript accordingly. I agree to its publication.